# m^6^A Methylation in Regulation of Antiviral Innate Immunity

**DOI:** 10.3390/v16040601

**Published:** 2024-04-13

**Authors:** Ivan Karandashov, Artyom Kachanov, Maria Dukich, Natalia Ponomareva, Sergey Brezgin, Alexander Lukashev, Vadim S. Pokrovsky, Vladimir Chulanov, Anastasiya Kostyusheva, Dmitry Kostyushev

**Affiliations:** 1Laboratory of Genetic Technologies, Martsinovsky Institute of Medical Parasitology, Tropical and Vector-Borne Diseases, First Moscow State Medical University (Sechenov University), 119048 Moscow, Russia; ivan.karandashov@gmail.com (I.K.); kachanov.av99@gmail.com (A.K.); dukich1999@mail.ru (M.D.); ponomareva.n.i13@yandex.ru (N.P.); seegez@mail.ru (S.B.); alexander_lukashev@hotmail.com (A.L.); 2Faculty of Virology, Lomonosov Moscow State University, 119234 Moscow, Russia; 3Division of Biotechnology, Sirius University of Science and Technology, 354340 Sochi, Russia; 4Department of Pharmaceutical and Toxicological Chemistry, Sechenov First Moscow State Medical University, 119048 Moscow, Russia; 5Shemyakin-Ovchinnikov Institute of Bioorganic Chemistry, Russian Academy of Sciences, 117997 Moscow, Russia; vadimpokrovsky@yandex.ru; 6Blokhin National Medical Research Center of Oncology, 117198 Moscow, Russia; 7Faculty of Biochemistry, RUDN University, 117198 Moscow, Russia; 8Department of Infectious Diseases, First Moscow State Medical University (Sechenov University), 119048 Moscow, Russia; vladimir@chulanov.ru; 9Faculty of Bioengineering and Biotechnologies, Lomonosov Moscow State University, 119234 Moscow, Russia

**Keywords:** TLR, RLR, herpesviruses, hepatitis viruses, coronaviruses, retroviruses, flaviviruses, adenoviruses

## Abstract

The epitranscriptomic modification m^6^A is a prevalent RNA modification that plays a crucial role in the regulation of various aspects of RNA metabolism. It has been found to be involved in a wide range of physiological processes and disease states. Of particular interest is the role of m^6^A machinery and modifications in viral infections, serving as an evolutionary marker for distinguishing between self and non-self entities. In this review article, we present a comprehensive overview of the epitranscriptomic modification m^6^A and its implications for the interplay between viruses and their host, focusing on immune responses and viral replication. We outline future research directions that highlight the role of m^6^A in viral nucleic acid recognition, initiation of antiviral immune responses, and modulation of antiviral signaling pathways. Additionally, we discuss the potential of m^6^A as a prognostic biomarker and a target for therapeutic interventions in viral infections.

## 1. Introduction

In 1974, Desrosiers, Friderici, and Rottman described a new modification of RNA, N6-methyladenosine (m^6^A), in Novikov’s hepatoma cells [1]. m^6^A is a post-transcriptional RNA modification involving the addition of an extra methyl group to nitrogen 6 of adenosine. Currently, more than 150 types of RNA modifications are identified [2], of which m^6^A is the most prevalent (Figure 1) [3].

The m^6^A modification has been observed in almost all types of RNA, including messenger RNA (mRNA) [4], ribosomal RNA (rRNA) [5], and microRNA [6]. m^6^A typically occurs at the consensus motif RRm^6^ACH ([G/A/U][G/A]m6AC[U/A/C]). Ke et al. demonstrated that 93% of m^6^A modifications in partially spliced chromatin-associated RNA were found within exonic regions, although intronic sequences are three-fold more abundant [7]. In addition, Dominissi et al. showed that 87% of m^6^A were located in exons longer than 400 nucleotides. Most m^6^A markers (37%) were found in coding sequences; 28% were localized in the 400-nucleotide window centered on stop codons; 20% were located in the 3′ untranslated region (3′-UTR); and 12% in transcription start sites (TSS). The relative enrichment of m^6^A bases was highest in the stop codon region and TSS [8].

The m^6^A modification plays a role in almost all biochemical processes related to RNA metabolism, influencing RNA stability [9] and regulating nuclear export of mRNA [10], splicing [11], and translation [12].

Viral RNA can also undergo methylation [13], and m^6^A influences various processes that regulate the viral cycle [14]. Thanks to m^6^A markers, cells can recognize their RNA as “self” and protect it from the innate immune system, whereas the genetic material of viruses can frequently be recognized as foreign because it does not bear m^6^A-modified nucleotides [15]. However, viruses have learned to evade cellular immunity by utilizing m^6^A modifications [16].

In this manuscript, we provide a focused review of the role of m^6^A in regulating innate immunity and shaping antiviral defenses. We consolidate recent results demonstrating the role of m^6^A methylation in viral immune evasion and antiviral immune signaling.

## 2. Regulation of m6A Modification

The m^6^A modification is dynamically reversible by three groups of enzymes: methyltransferases (writers), demethylases (erasers), and proteins that recognize m^6^A-modified RNA (readers). The reversibility of m^6^A methylation is controlled by writers and erasers, while reader proteins recognize the modified adenosines and regulate the associated biological functions (Figure 2). 

### 2.1. Writers

So far, four main genes encoding methyltransferases have been identified in the human genome—methyltransferase-like (METTL) 3 (METTL3), METTL5, METTL16, and zinc finger CCHC-type containing 4 (ZCCHC4). In addition to these methyltransferases, other proteins participating in m^6^A catalysis perform collaborative functions, including METTL14, Wilms tumor 1-associating protein (WTAP), Vir-like m^6^A methyltransferase associated (VIRMA), Cbl proto-oncogene like 1 (HAKAI), zinc finger CCCH-type containing 13 (ZC3H13), and RNA-binding motif 15/15B (RBM15/15B). In most cases, m^6^A methylation is catalyzed by the methyltransferase complex (MTC) consisting of METTL3, METTL14, WTAP, RBM15/15B, ZC3H13, VIRMA, and HAKAI, which is typically localized in the nucleus, except for in certain cancer cell lines [17,18]. METTL3 directly adds m^6^A modifications onto RNA, and METTL14 stabilizes the conformation of METTL3’s catalytic center. METTL14 is also crucial for substrate recognition [19]. In addition, Liu et al. have also identified that both METTL14 and METTL3 individually exhibit methyltransferase activity; however, the activity of the METTL3–METTL14 complex is markedly higher [20]. The remaining proteins of this complex perform various specific functions: WTAP targets the METTL3–METTL14 complex to nuclear speckles [17,21], and RBM15/15B binds to the m^6^A complex and interacts with specific U-rich sites in XIST RNA. Data show that knockdown of both RBM15 and RBM15B results in impaired XIST-mediated gene silencing, suggesting that RBM15 and RBM15B are necessary for MTC recruitment to XSIT RNA. Additionally, WTAP is essential for facilitating the interaction between RBM15/RBM15B and the methylation complex [22]. ZC3H13 regulates MTC’s localization to the nucleus, as knocking down ZC3H13 results in cytoplasmic localization of METTL3, METTL14, VIRMA, WTAP, and HAKAI, the latter of which bridges WTAP to the mRNA-binding factor NITO [23]. VIRMA also plays a crucial role in preferential 3′-UTR m^6^A modification, but has no impact on long-exon-preferential methylation [24]. Studies on *Drosophila melanogaster* have shown that HAKAI is crucial for MTC stabilization, as its knockout results in decreased methylation activity. We were unable to find studies regarding the role of HAKAI in mammals, but HAKAI is considered to be conserved between humans and *Drosophila* [25]. 

Methylation of 18S and 28S rRNA, as well as small nuclear RNA (snRNA) U6, is carried out by the heterodimeric methyltransferase complex METTL5 and the stabilizing coactivator protein TRMT112 [26]. m^6^A modification of 28S rRNA is regulated by methyltransferase ZCCHC4. ZCCHC4 is localized in nucleoli, where it interacts with RNA-binding proteins involved in ribosome biogenesis and RNA metabolism [27]. The methyltransferase METTL16 is distributed to both the nucleus and the cytoplasm [28,29], and interacts with snRNA U6, rRNA, and pre-mRNA [30,31]. Notably, METTL16 can regulate the expression of MAT2A, which is directly responsible for the synthesis of S-adenosylmethionine [32]. 

In addition to the main methyltransferases, m^6^A modification can be added on mRNA cap-adjacent Am-modified nucleotides by cap-specific adenosine-N^6^-MTase (CAPAM). Using RNA-MS, Akichika S. et al., demonstrated that CAPAM catalyzes N6-methylation of m^6^Am in capped mRNA in an S-adenosylmethionine (SAM)-dependent manner. Additionally, CAPAM specifically recognizes the cap structure 7-methylguanosine (m^7^G) and N^6^-methylates m^7^GpppAm [33].

### 2.2. Erasers

Protein erasers act as demethylases, removing m^6^A modifications from RNA. Currently, only two demethylases from the FeII/α-KG-dependent dioxygenase AlkB family associated with m^6^A methylation have been described. The first demethylase, identified in 2011, is the fat mass- and obesity-associated protein (FTO) [34], which regulates processing and alternative splicing in adipocytes through m^6^A demethylation [35]. Mathiyalagan et al. discovered that FTO-dependent demethylation of m^6^A regulates intracellular Ca^2+^ dynamics and sarcomeres in cardiomyocytes [36]. FTO’s activity as an m^6^A methyltransferase is interrelated with lipid accumulation control in muscle, as this enzyme plays a regulatory role in activating AMP kinase (AMPK) [37]. Wu et al. demonstrated that suppressing FTO significantly reduces the expression of Cyclin A2 (CCNA2) and cyclin-dependent kinase 2 (CDK2) genes, which play a key role in cell cycle regulation. This leads to the delayed transitioning of cells exposed to insulin into the G2 phase of the cell cycle. Moreover, the level of m^6^A methylation of CCNA2 and CDK2 mRNA is substantially increased upon FTO suppression [38]. 

Another demethylase is the related alpha-ketoglutarate-dependent dioxygenase alkB homolog 5 (ALKBH5), described in 2013 [10]. ALKBH5 primarily interacts with m^6^A in RNA and is most highly expressed in the testes and lungs. ALKBH5 is believed to regulate the assembly of mRNA processing factors [10]. Notably, m^6^A sites are the only substrates for ALKBH5, whereas FTO can erase other modifications like 2-O-dimethyladenosine, N1-methyladenosine, 3-methylthymine, and 3-methyluracil [39,40].

### 2.3. Readers

Reader proteins are the major factors that execute biological functions in m^6^A-modified RNA by recognizing and binding to methylated transcripts. Three major groups of reader proteins regulate m^6^A-mediated functions. The first group is proteins containing the conservative YT521-B homology (YTH) domain [41], which consists of approximately 150 amino acids distributed across three α-helices and six β-sheets [42]. In humans, five proteins with the YTH domain have been identified: three paralogs of the YTH domain family 1–3 (YTHDF1, YTHDF2, and YTHDF3) and distinct proteins of the YTH domain containing 1–2 (YTHDC1 and YTHDC2). All of these proteins have structural differences. In particular, YTHDC1 contains the YTH domain surrounded by charged regions containing glutamic acid, arginine, or proline residues. YTHDC2 is the most complex protein in this group, containing not only the YTH domain, but also several helicase domains and two ankyrin repeats, possessing ATPase and 3′→5′-helicase activity [43]. The YTHDF family of proteins, together with the YTH domain, also consist of disordered regions enriched in proline, glutamine, and asparagine [42]. 

YTHDF1–3 and YTHDC2 mainly localize in the cytoplasm, while YTHDC1 is predominantly distributed in the nucleus. YTHDF1 enhances the translation of m^6^A-modified RNAs by a yet unclarified mechanism; YTHDF1 binds RNA near the stop codon and interacts with translation initiation factor complex 3, which is part of the translation initiation complex [12]. YTHDF2’s functions are primarily associated with mRNA degradation in the cytosol. It can activate the carbon catabolite repression 4-negative on the TATA-less (CCR4-NOT) deadenylase complex, involved in mRNA deadenylation and degradation [44], and the P/MRP ribonuclease complex, which initiates endoribonucleolytic cleavage of mRNA. YTHDF3 is the least studied of this group. Li et al. noted that YTHDF3 regulates translation by interacting with YTHDF1 and with the 40S and 60S ribosome subunits [45]. However, it has also been determined that YTHDF3, along with YTHDF2, can participate in degrading RNA [46]. YTHDC1 actively participates in the regulation of transcription, splicing, and RNA export from the nucleus [47], while YTHDC2 contributes to enhancing translation efficiency and mRNA degradation through its helicase activity [48].

The second group of readers includes several heterogeneous nuclear ribonucleoproteins (HNRNP): HNRNPC, HNRNPG, and HNRNPA2B1. HNRNPC selectively binds to unstructured RNA regions during pre-mRNA processing and separates transcripts into mRNA and uridine-rich snRNA [49]. HNRNPG binds to arginine-glycine-glycine (RGG) regions and regulates alternative splicing of pre-mRNA by interacting with the phosphorylated C-terminal domain of RNA polymerase II [50]. HNRNPA2B1 accelerates the processing of primary microRNAs by interacting with the microprocessor complex subunit DGCR8 [51].

The third group of readers consists of three highly conserved insulin-like growth factor 2 mRNA-binding proteins (IGF2BP): IGF2BP1, IGF2BP2, and IGF2BP3. At the N-terminus, IGF2BP proteins have two RNA recognition motifs (RRM); four hnRNP-K homology (KH) domains are located at the C-terminus [52]. KH domains are mainly responsible for protein–RNA binding. RRM domains regulate the stability of IGF2BP–RNA complexes [53]. The primary function of IGF2BP proteins is maintaining stability of target RNAs [54]. It is suggested that virtually any cellular protein can act as reader for m^6^A, enabling highly tunable and diverse control of RNA metabolism [55]. 

### 2.4. m^6^A Modifications and Viruses

m^6^A methylation has been shown to influence the life cycle of viruses by affecting viral replication and evasion of the innate immune response. To date, there is a large body of research focused on the role of m^6^A in regulating replication of viruses from many different families. For example, Lu W et al. demonstrated that m^6^A modification at two sites in the 5′ leader sequence of HIV-1 genomic RNA (gRNA) promotes viral replication. Mutations in these sites resulted in HIV-1 with lower infectivity compared to wild-type virus. Additionally, overexpression of m^6^A reader proteins YTHDF1-3 in target cells reduces the levels of HIV-1 gRNA, inhibiting early and late reverse transcription processes [56]. 

Ye et al., revealed that Kaposi’s sarcoma-associated herpesvirus (KSHV) can also manipulate the host m^6^A machinery to its own advantage by switching from latency to lytic infection. It was demonstrated that most transcripts of KSHV and their levels increase upon lytic infection. This effect was mediated by the interaction of m^6^A sites with YTHDC1 reader proteins in conjunction with two splicing factors: serine/arginine-rich splicing factor 3 (SRSF3) and SRSF10. Mutating m^6^A sites blocked splicing of pre-mRNA (RTA—replication transcription activator) that encodes a key KSHV protein, required for lytic infection. Therefore, m^6^A modification plays an important role in viral splicing [57]. Similarly, m^6^A modification accelerates replication of enterovirus 71 (EV71) due to complex interactions between m^6^A-modified viral RNA, m^6^A machinery, and RNA-dependent RNA polymerase 3D [58]. These and other effects of m^6^A modifications in viral RNA on viral life cycle are discussed in detail in the next chapters. 

## 3. Mechanisms of the Antiviral Innate Immune Response

Organisms are constantly exposed to exogenous antigens and patterns, including pathogenic microorganisms like bacteria and viruses. To eliminate such pathogens and maintain the host’s health, two interrelated immune systems have evolved: innate and acquired immunity. The immune system is tightly regulated to avoid the risk of non-specific recognition of self-antigens, which can not only impair the ability to eliminate pathogens but also negatively impact the host. Therefore, cells possess various regulatory mechanisms for both innate immune perception and the transmission of signals that initiate antimicrobial response reactions (Figure 3).

### 3.1. Signaling Cascades of Innate Immunity

Recognition of nucleic acids is one strategy by which cells can detect infectious agents. In recent years, tremendous progress has been made in understanding how cells can activate the immune response at the molecular level. The binding of intracellular nucleic acids to a range of specialized sensors activates downstream signaling cascades, leading to the production of type I interferons (IFN) and pro-inflammatory cytokines, triggering corresponding systemic immune responses.

After cells are infected by a virus, the innate immune response is triggered by proteins called pathogen recognition receptors (PRRs), that recognize specific molecular structures of pathogens and play a crucial role in innate immunity by activating the immune response upon contact with pathogens. Currently, five subfamilies of PRRs have been described: membrane Toll-like receptors (TLRs), RIG-I-like receptors (RLRs), Nod-like receptors (NLRs), AIM2-like receptors (ALRs), and C-type lectin receptors (CLRs) [59]. In most cases, membrane PRRs are actively expressed in immune cells, such as macrophages or dendritic cells (DCs). Intracellular receptors actively participate in processes related to the activation of the immune response, including apoptosis, phagocytosis, and the regulation of gene activity.

PRRs can interact with pathogen-associated molecular patterns (PAMPs) and damage-associated molecular patterns (DAMPs). In addition to intracellular receptors, viral genetic material in the cytoplasm can be identified by several factors, the most common of which are cyclic GMP-AMP synthase (cGAS), interferon-inducible protein 16 (IFI16), and absent in melanoma 2 (AIM2).

PAMPs are molecular patterns that consist of conserved microbial and viral components (nucleic acids, proteins, and carbohydrates) [60], and DAMPs are molecules released during cellular stress or tissue damage. These molecules act as endogenous danger signals, triggering inflammatory reactions and activating the innate immune system during non-infectious inflammatory processes [61].

Three main classes of PRRs can interact with viral genetic material: RLR [62], TLR [63], and a set of cytosolic sensors [64,65,66]. RLRs interact with viral RNA, which is transcribed from RNA and DNA viruses. TLRs can directly interact with various substrates, such as double-stranded RNA (dsRNA; TLR-3), single-stranded RNA (ssRNA; TLR-7/TLR-8), and CpG DNA (TLR-9) in endolysosomes upon viral entry. 

Recognition of foreign nucleic acids by PRRs or DNA sensors triggers the activation of adapter proteins that are crucial components of the signaling pathways governing the production of IFNs and pro-inflammatory cytokines. Activation of TLR3 results in the activation of TRIF protein, which subsequently leads to the activation of TBK1 kinase. On the other hand, TLR8/9 interact with the MyD88 adapter, which triggers the activation of IRAK kinases. RIG-like receptors engage with the mitochondrial antiviral signaling (MAVS) adaptor, which is associated with the common mitochondrial membrane-linked MAVS adaptor to transmit antiviral signals through the TBK1 kinase pathway. DNA sensors, such as cGAS and IFI16, trigger the secretion of IFNs and cytokines by activating STING adaptor protein and promoting TBK1 activity. 

The signal transduction cascade involving IRAK and TBK1 adaptor kinases leads to the phosphorylation of IRF transcription factors, including IRF3, IRF7, and IRF5. This phosphorylation event facilitates the homodimerization of IRFs, their translocation to the nucleus, and interaction with promoters of IFN I and IFN III genes.

An alternative signaling pathway involving IRAK and TBK1 leads to the phosphorylation of the IκB-α protein. This phosphorylation event triggers the degradation of IκB-α, resulting in the release and activation of NF-κB. Activated NF-κB then translocates to the nucleus and interacts with NF-κB-dependent promoters of pro-inflammatory cytokines. The inflammatory response is modulated by proinflammatory cytokines such as TNF, IL-1, and IL-6 [67].

The antiviral effects of type I and III IFNs are mediated through the activation of interferon-stimulated genes (ISGs). Upon the binding of IFN molecules to their respective receptors, which consist of IFNAR1/IFNAR2 proteins for type I IFN and IFNLR1/IL-10R2 for type III IFN, adaptor kinases JAK/TYK are activated. This activation leads to the phosphorylation of STAT1/2 transcription factors. The phosphorylated STAT1/2 proteins form a heterotrimeric activation complex known as interferon-stimulated gene factor 3 (ISGF3) with IRF9. This complex is then translocated to the nucleus where it binds to Interferon-sensitive response element (ISRE) consensus motifs in the promoters of ISGs, thereby inducing their expression (Figure 4).

After pathogen recognition, a cascade of protein reactions is initiated, leading to the production of various protective molecules in the organism, including IFNs, pro-inflammatory cytokines, and chemokines [67].

### 3.2. Recognition of Viruses by RLR

Three proteins with DExD/H-box helicase activity belong to the RLR subfamily: RIG-I, melanoma differentiation association gene 5 (MDA5), MAVS, and laboratory of genetics and physiology 2 (LGP2). RLRs include a central DExD/H-box helicase/ATPase domain and a C-terminal regulatory domain that binds to RNA and zinc ligands. RIG-I and MDA5 have two tandem caspase recruitment domains (CARDs) at their N-terminus [68,69]. RIG-I recognizes relatively short ssRNA and dsRNA (up to 1 kb) with triphosphate or diphosphate fragments at the 5′-end [70]. In contrast, MDA5 detects longer dsRNA (over 1 kb) formed during viral replication [71]. Further investigation is warranted to elucidate the function of LGP2, though LGP2 can interact with viral dsDNA and ssRNA, modulating the functions of RIG-I and MDA5. Conflicting findings suggest that LGP2 may exert a dual role in regulating RIG-I and MDA5 activity, either negatively or positively influencing MDA5 [65]. LGP2 has been observed to enhance the action of MDA5 [72], but can inhibit the action of the TRAF ubiquitin ligase, thereby negatively regulating the innate immune response [73].

RLRs are expressed in virtually all tissues. Upon binding to a virus, RIG-I and MDA5 undergo a conformational change, exposing the CARD domain [74,75], which binds to the CARD domain at the N-terminus MAVS. MAVS is anchored to the outer membrane of the mitochondria through the C-terminal transmembrane domain, and forms functional aggregates upon activation [76]. This complex includes TNF receptor-associated factors (TRAF) and kinases TBK1 and IκB-ε (IKKe). The complex also consists of subunits IKKα, IKKβ, and IKKγ of the tripartite-activated protein kinase (TAK1), which subsequently stimulates the activity of interferon regulatory factor 3 (IRF3) and/or IRF7, as well as nuclear factor kappa B (NF-κB) [77]. This cascade stimulates the production and release of IFNs, cytokines, and IFN-stimulated genes (ISGs). In addition to mitochondria, MAVS is found in peroxisomes, where it activates the expression of IFN III genes [78].

### 3.3. Recognition of Viral Nucleic Acids by TLRs 

TLRs are located on the cell surface or inside the cell in organelles such as the endoplasmic reticulum (ER), endosomes, lysosomes, or endolysosomes. TLR synthesis occurs in the ER. Subsequently, TLRs exit the ER with the help of the protein UNC93B1 and move into endosomes through the plasma membrane or directly through the Golgi complex [63,79]. Each TLR molecule has an N-terminal domain with leucine-rich repeats (LRR), a transmembrane domain, and a Toll/IL-1 receptor (TIR) cytoplasmic domain at the C-terminus. The LRR domain recognizes PAMPs, and the TIR domain activates downstream signaling pathways. Viral nucleic acids are typically recognized by intracellular TLR3, TLR7, TLR8, and TLR9. TLR3 recognizes viral dsRNA, TLR7 and TLR8 recognize ssRNA, and TLR9 recognizes the unmethylated oligodeoxynucleotide CpG [80].

The cellular antiviral response mediated by TLR is based on the recruitment of adapters containing the TIR domain (MyD88, TRIF, TIRAP/MAL, or TRAM). The TLR signaling pathway can be broadly divided into two pathways: MyD88-dependent and TRIF-dependent [81]. In the MyD88-dependent pathway, either the IKK complex or the MAPK pathway can be activated. The IKK complex releases NF-κB, which translocates to the nucleus and activates the expression of pro-inflammatory cytokine genes. The MAP kinase cascade is responsible for the formation of the transcription factor AP-1, which is also directed at cytokine genes. The TRIF-dependent pathway interacts with TRAF6 and TRAF3. In turn, TRAF6 activates the NF-κB signaling cascade and, consequently, the production of cytokines. TRAF3 induces the expression of IFN-I genes.

### 3.4. Recognition of Viruses Using Cytosolic Sensors

Viral DNA present in the cytosol of eukaryotic cells is also a PAMP. Cytoplasmic viral DNA can be recognized by a set of sensors like cGAS, IFI16, and AIM2. Additionally, several other proteins like DEAD-box RNA helicase 41 (DDX41) and RNA polymerase III have been noted to initiate innate immune responses in cells [82]. cGAS binds to the sugar-phosphate backbone of dsDNA without sequence specificity, allowing it to recognize a wide variety of DNA types. Due to this property, cGAS can detect numerous DNA species. The C-terminal domain of cGAS includes a nucleotidyltransferase. Upon binding to viral DNA, cGAS catalyzes the production of cGAMP from ATP and GTP. cGAMP, in turn, acts as a secondary messenger and can activate stimulator of IFN genes (STING) [83,84].

Another sensor of viral DNA in the cytosol is IFI16, a member of the pyrin and hematopoietic interferon-inducible nuclear (HIN) domain (PYHIN) protein family. IFI16 is predominantly localized in the nucleus but can shuttle between the nucleus and the cytoplasm. IFI16 contains a pyrin domain at the N-terminus and two HIN200 domains at the C-terminus. It interacts with viral DNA through the HIN200 domain, after which IFI16 interacts with cGAS, initiating cGAMP production [85].

Also belonging to the PYHIN family is the AIM2 protein, which also recognizes viral DNA. Like IFI16, AIM2 consists of two domains: a pyrin domain at the N-terminus and an HIN200 domain at the C-terminus. The HIN200 domain is responsible for DNA binding, while the pyrin domain interacts with the pyrin domain of the adapter molecule apoptosis-associated speck-like protein containing a caspase recruitment domain (CARD) (ASC). AIM2 initiates caspase 1-dependent activation of inflammation, leading to the production of interleukin-1β (IL-1β) and IL-18 [86].

Other sensors of viral DNA and RNA have been described in the cytosol, including DDX41 [87], RNA polymerase III [88], DNA-dependent protein kinase (DNA-PK) [89], oligoadenylate synthase (OAS) [90], and many other factors. However, there are still numerous questions regarding the role of these factors and intracellular immune signaling pathways, as well as the ways in which viruses evade detection. All these questions require further detailed description and study.

A key player in the activation of cellular immunity by the aforementioned agents is STING, an ER membrane protein that consists of a cytosolic N-terminal domain, four transmembrane helices forming the transmembrane domain, and a cytosolic ligand-binding domain (LBD) to which the C-terminal domain is attached. In cells, STING exists as a V-shaped dimer [91]. The LBD binds to cGAMP [92], and the C-terminal tail contains the PXPLRXD motif and is necessary for the activation of TANK-binding kinase 1 (TBK1) [93]. After activation, TBK1 phosphorylates IRF3, as mentioned earlier. Consequently, IRF3 translocates to the nucleus and induces the synthesis of anti-inflammatory cytokines and type I IFNs [94].

Numerous studies analyzed the intricate mechanisms underlying TLRs functions, including their interactions with nucleic acids and antiviral activities. These interactions were reviewed previously [95,96].

### 3.5. Recognition of Viruses in the Nucleus

Usually, PRRs are localized on the plasma membrane, in endosomes, or in the cytoplasm. However, some nuclear proteins can also serve as viral sensors. For example, the aforementioned IFI16, cGAS, and hnRNPA2B1 recognize herpes simplex virus 1 (HSV1) in the nucleus [97,98,99]. In uninfected cells, hnRNPA2B1 is methylated. Upon HSV1 infection, hnRNPA2B1 forms a complex with viral DNA, dimerizes, and is demethylated by Jumonji domain-containing 6 (JMJD6). This results in its cytoplasmic translocation followed by activation of TBK1, enhanced phosphorylation of IRF3, and activation of the immune response mediated by IFN I signaling (Figure 5) [100]. Additionally, Carpenter et al. demonstrated that hnRNPA2B1 factor binds HSV1 DNA in the nucleus, amplifying IFNβ antiviral signaling (Figure 5) [99]. 

Gentili et al. found that nuclear cGAS enhances innate immune responses. Associated with centromeres and DNA-repetitive sequences, nuclear cGAS can synthesize cGAMP and stimulate innate immune activity in primary DCs [101]. Nuclear cGAS can also bind to RNA viruses. In the nucleus, cGAS interacts with PRMT5 and facilitates symmetric demethylation of histone H3 arginine 2 at the IFN-I promoter element, thus promoting interaction of activated IRF3 with this promoter and enhancing production of IFN I and C-X-C motif chemokine ligand 10 (CXCL10) [102].

Similar to its role as a cytoplasmic viral sensor, IFI16 functions as a viral sensor in the nucleus. Upon binding to viral DNA, IFI16 activates the nuclear protein DNA-PK, which, in turn, phosphorylates IFI16 at T149 [103]. This modification determines IFI16’s subcellular localization and protein nuclear export, and promotes synthesis of type I IFNs. IFI16 undergoes a conformational change to adopt a filamentous structure during the course of infection, leading to its localization to viral replication sites [104]. After exiting to the cytoplasm, IFI16 can bind and activate STING [105].

As shown by Diner et al., IFIX, another protein of the PYHIN family, also acts as a viral sensor in the nucleus. The authors noted that upon IFIX overexpression, HSV1 titers decreased almost three-fold, while IFIX knockdown significantly increased viral titers, suggesting that IFIX is an antiviral factor. Crow and Cristea discovered that HSV-1 has acquired mechanisms to block IFIX function via proteasome-dependent degradation of the pyrin domain. Using immunoprecipitation–mass spectrometry (IP-MS), the authors demonstrated that IFIX interacts with components of the ubiquitin–proteasome system and transcriptional regulators during HSV-1 infection [106]. 

A nuclear sensor for viral dsRNA is scaffold attachment factor A (SAFA). SAFA interacts with dsRNA, oligomerizes, and activates DNA topoisomerase 1 and the SWI/SNF-related matrix-associated actin-dependent regulator of chromatin subfamily A member 5, which, in turn, regulate the synthesis of enhancers for antiviral genes, including IFN-β1 [107].

Other note-worthy regulators are the non-POU domain-containing octamer binding (NONO) and hexamethylene bis-acetamide-inducible protein 1 (HEXIM1). NONO has been found to recognize the conserved capsid region of human immunodeficiency virus (HIV) and binds to cGAS [108]. HEXIM1 binds to the long non-coding RNA (lncRNA) NEAT1 and forms a complex that activates the cGAS–STING pathway [109].

## 4. m^6^A Methylation of Antiviral Response Factors

One important function of m^6^A modification is the activation and regulation of intracellular immunity. An early study showed that depleting METTL14 induces synthesis of type I IFNs, while depleting the demethylase ALKBH5 reduces type I IFN expression [110]. Further, m^6^A was found to be responsible for complex regulation of distinct antiviral signaling programs. 

### 4.1. m^6^A-Dependent Regulation of the RLR Signaling Pathway

#### 4.1.1. m^6^A and Stimulation of Immunity

m^6^A modification plays a crucial role in the RLR-dependent antiviral response pathway. m^6^A methylation is necessary for the export of MAVS, TRAF3, and TRAF6 mRNA factors from the nucleus. Zheng et al. found that during viral infection, the eraser protein ALKBH5 is activated, preventing the export of these transcripts from the nucleus and thus inhibiting IFN synthesis (Figure 6) [111]. Hesser and Walsh demonstrated that the expression of YTHDF readers was suppressed to varying degrees during infection with cowpox virus (VacV) and HSV1. The authors noted that decreased levels of YTHDF1, YTHDF2, and, to a lesser extent, YTHDF3, led to increased expression of ISGs and activation of antiviral response mechanisms [112], indicating the crucial role of readers in shaping the innate immune response. Knocking out METTL14 in macrophages infected with Sendai virus or encephalomyocarditis virus, Qin et al. found that demethylation of MAVS, which regulates the RLR-induced signaling pathway, enhanced expression, preserved protein stability, and consequently amplified IFN synthesis [113]. The authors also observed a gradual increase in the expression of eraser proteins FTO and ALKBH5 during infection with these viruses. Further research is needed to investigate the possibility of demethylation by eraser proteins and the role of demethylated MAVS.

#### 4.1.2. Viral Evasion of RLR-Mediated Immune Responses

Viruses have developed strategies to evade RLR-mediated immunity. According to Li et al., the 3′-UTR of the genome of severe acute respiratory syndrome coronavirus clade 2 (SARS-CoV-2) is enriched with m^6^A-modified nucleotides. SARS-CoV-2 utilizes the host’s METTL3 methyltransferase for innate immune response evasion. Depleting METTL3 reduced m^6^A methylation in the SARS-CoV-2 genome and resulted in decreased viral RNA levels. It also led to reduced m^6^A methylation and expression of host proviral genes, including but not limited to neuropilin-1 (NRP1), tripartite motif containing 4 (TRIM4), and Suppressor of Mothers against Decapentaplegic (SMAD) family member 4 (SMAD4). In contrast, expression of innate immune response effector genes was upregulated in METTL3- and METTL14-depleted cells. This upregulation was due to the increased binding of m^6^A-unmethylated viral RNA to RIG-I, which was further confirmed by introducing mutations into SARS-CoV-2 m^6^A sites [114]. 

Qiu et al. found that METTL3 is a strong internal inhibitor of the innate immune response in vascular stomatitis virus (VSV)-infected cells, suppressing innate immunity signal transduction and inhibiting IFN synthesis. VSV infection robustly enhances translocation of METTL3 but not METTL14 into the cytoplasm. Using PAR-CLIP-seq and miCLIP-seq, the group identified METTL3 binding sites on positive-sense RNAs of VSV, unlike negative-sense genomic RNA. m^6^A methylation of viral RNA impeded dsRNA formation, making it less prone to sensing by RIG-I and MDA5 and, subsequently, reducing IFNB-1 production (Figure 6) [115]. 

### 4.2. m^6^A-Dependent Regulation of the TLR Signaling Pathway

#### 4.2.1. m^6^A and Stimulation of Cellular Immunity through the TLR Pathway

m^6^A methylation can also influence the TLR-mediated signaling pathway. In their study of dental pulp inflammation, Feng et al. showed that knocking down methyltransferase METTL3 regulated alternative splicing of the TLR-mediated antiviral immune pathway participant MyD88, resulting in the production of a MyD88S splice variant, which is known to inhibit inflammatory cytokine production. Depletion of METTL3 reduced the accumulation of inflammatory cytokines and suppressed the activation of NF-κB and MAPK signaling pathways (Figure 7) [116]. Similarly, METTL3 overexpression in a mouse model of inflammatory bowel diseases resulted in increased immune response via the NF-κB pathway [117]. Using m^6^A-modified in vitro RNA oligonucleotides, Karikó et al. demonstrated that m^6^A modification reduced the ability of RNA to induce cytokine secretion mediated by TLR3, TLR7, and TLR8 [118], indicating the crucial role of this methylation in pathogen recognition and differentiation between pathogen and host nucleic acids. Tong et al. used CRISPR tools to demonstrate that macrophages with METTL3 deficiency exhibit reduced activation associated with TLR signaling. Decreased methylation of IRAKM factor led to an increased level of IRAKM, which inhibited signal transduction in TLR-dependent macrophage activation [119]. Additionally, m^6^A modification of CD40, CD80, and toll/interleukin-1 receptor domain-containing adapter protein (TIRAP) transcripts promoted DC activation and DC-based immune response activation of TLR [120]. Notably, experiments by Geng et al. on brown croaker fish (*Miichthys miiuy*) showed that, in cells infected with *Siniperca chuatsi* rhabdovirus and *Vibrio anguillarum* bacteria, METTL3 inhibited the innate immune response via methylation of TRIF and MYD88, mRNA factors in the TLR pathway that are further degraded through YTHDF2- and YTHDF3-dependent mechanisms. In contrast, YTHDF1 promoted the translation of MYD88 RNA [121]. 

#### 4.2.2. m^6^A-Dependent Regulation of Cytosolic and Nuclear Sensors

m^6^A methylation plays a crucial role in recognizing viruses in the nucleus. As previously mentioned, one nuclear sensor is the reader protein hnRNPA2B1. Wang et al. showed that, after binding to HSV-1 DNA in the nucleus, hnRNPA2B1 is demethylated at arginine-226 by the arginine demethylase JMJD6, after which it exits to the cytoplasm and activates the TBK1–IRF3 pathway in a STING-dependent manner. In vitro knockdown of hnRNPA2B1 or in vivo knockout of this factor in mice reduced the expression of IFN-I induced by DNA virus HSV-1; no such reduction was seen in RNA viruses. hnRNPA2B1 also promotes m^6^A modification and translocation to the cytoplasm of cGAS, STING, and IFI16 mRNA, enhancing their expression and ensuring a robust immune response (110). Balzarolo et al. identified that m^6^A modification augments recognition of cytosolic dsDNA through the cGAS–STING pathway. m^6^A methylation of 5′-GATC-3′ motifs increases the immunogenicity of synthetic dsDNA in mouse macrophages and DC [122]. Song et al. showed that m^6^A can regulate lncRNAs associated with the cGAS–STING pathway, stimulating several key components of the cGAS–STING signaling pathway, including STAT1, DNA sensors cGAS and hnRNPA2B1, and the IFN receptor IFNAR1 [123].

### 4.3. m^6^A-Dependent Regulation of Type I Interferon Synthesis

McFadden et al. conducted an extensive study and demonstrated that the MTC complex methylates a series of IFN-stimulated genes. Depletion of the METTL3/METTL14 complex resulted in decreased expression of ISGs IFITM1 and MX1 while the expression of ISG15 and EIF2AK2 remained unchanged. m^6^A modification in the 3′-UTR of IFITM1 enhances its translation through interaction with YTHDF1. The authors also noted that knockdown and overexpression of METTL3/14 proportionally altered the expression of type I IFN, but did not alter the expression of MX1, ISG15, or EIF2AK2 [14]. Ge et al. observed that, in cells with WTAP knockdown, the expression of phosphorylated and total IRF3, which affects IFN synthesis, was reduced, and its dimerization was disrupted. In addition to IRF3, WTAP degradation led to decreased levels of IFNAR1, another key factor in IFN-I signaling. The authors also noted that in peripheral blood mononuclear cells (PBMC) infected with vesicular stomatitis virus with enhanced GFP, HSV1, or intracellular HSV-60 (a synthetic analog of HSV1 DNA), WTAP was degraded via the ubiquitination-proteasome pathway due to activation of IFN-I signaling. WTAP degradation resulted in reduced m^6^A methylation of IRF3 and IFNAR1 mRNA, resulting in reduced IRF3 and IFNAR1 protein levels and blocking of IFN-1-mediated innate response. The conclusion was that WTAP–IRF3/IFNAR1 signaling may serve as a negative feedback pathway of INF-I signaling [124].

Imam et al. found that m^6^A-methylated hepatitis B virus (HBV) RNA is degraded by ISG20 via YTHDF2 m^6^A reader recruitment. They found that YTHDF2 protein was crucial for recognizing m^6^A-methylated viral RNA. After recognition, YTHDF2 recruited ISG20 to facilitate viral RNA degradation [125]. Kim et al. also reported that HBV expression induces m^6^A modification of phosphatase and tensin homolog (PTEN) mRNA, thus reducing PTEN RNA stability mainly through interaction with YTHDF2 and YTHDF3. PTEN increases nuclear import of IRF-3 and promotes subsequent IFN synthesis; thus, decreased PTEN expression due to the lower stability of m^6^A-methylated mRNA hinders IFN production. Data from patients with hepatocellular carcinoma (HCC), HBV-positive patients with HCC, and patients negative for both HBV and HCC also suggest that HBV-induced m^6^A-methylation of PTEN might be responsible for HCC development via disruption of the PI3K pathway (Figure 8) [126].

Similarly, Zhang et al. showed that knocking down the reader protein YTHDF3 reduced the replication of VSV, HSV1, and encephalomyocarditis virus in infected cells. Subsequently, the authors used RNA sequencing to demonstrate that YTHDF3 knockout cells exhibited higher expression of various ISGs, including IFIT1, CXCL10, MX1, and OAS1a. However, it was noted that YTHDF3 did not bind to ISGs or signaling factors of the JAK–STAT pathway and did not affect the stability of ISG mRNA. According to eCLIP-seq data, even basal levels of YTHDF3 could inhibit IFN synthesis by promoting forkhead box O3 (FOXO3) protein translation. FOXO3 is known to negatively regulate ISG expression. YTHDF3 binds FOXO3 mRNA independently of METTL3-mediated m^6^A modifications and promotes its translation through interaction with poly-A binding protein and eukaryotic translation initiation factor 4 gamma 2 [127].

## 5. The Role of the m^6^A Modifications in the Genetic Material of Viruses

In addition to its involvement in cellular processes, the RNA modification has been shown to impact the regulation of the viral life cycle. There is substantial evidence that modifications other than m^6^A can affect the replication of various viruses such as 5-methylcytidine (m^5^C) [128], N4-acetylcytidine (ac^4^C) [129], and 2′O-methylation of the ribose fragment of all four ribonucleosides (Nm), etc. [130]. However, m^6^A modifications are the most prevalent. The presence of m^6^A markers in viruses has been recognized for several decades, but its functional significance has been poorly understood until just recently. Initially identified in nuclear viral DNA, the role of m^6^A was later described for RNA viruses. Upon infection, writer and eraser proteins can translocate from the nucleus to the cytoplasm, where they regulate methylation of viral nucleic acids. Further investigation of m^6^A methylation in viruses is crucial for understanding how it influences the viral epitranscriptome and may provide insights for developing novel prognostic biomarkers and antiviral therapies.

### 5.1. Retroviruses 

HIV-1, a representative of the *Retroviridae* family, is a positive ssRNA virus that replicates in the nucleus and is susceptible to m^6^A methylation. Tirumuru et al. demonstrated that METTL3 and METTL14 knockdown resulted in inhibited HIV-1 Gag protein expression. Conversely, knocking down ALKBH5 and FTO resulted in higher HIV protein expression; notably, FTO knockdown caused a significant increase in HIV protein expression. m^6^A modifications in the HIV-1 genome can interact with cellular YTHDF1–3 reader proteins. Partially knocking down YTHDF1 or YTHDF3 in Jurkat cells resulted in a three- to four-fold increase in HIV-1 infection, while only a slight increase was observed after YTHDF2 knockdown. Overexpressing each of the YTHDF1–3 proteins significantly reduced the level of HIV-1 reverse transcriptase products by four- to ten-fold. This downregulation of reverse transcription was attributed to mRNA degradation, mediated by YTHDF1–3 [131]. In contrast, in the study by Tsai et al., a three-fold increase was observed in the overall expression of HIV-1 viral RNA in cells with YTHDF2 protein overexpression 24 h after infection. The authors noted that YTHDF2 and another reader, YTHDC1, bind to m^6^A sites in HIV-1 transcripts. Knockdown of YTHDC1 increased viral Gag expression, while overexpressing YTHDC1 reduced viral RNA expression. Deep sequencing of splice forms demonstrated that YTHDC1 can regulate alternative splicing of HIV-1 transcripts [132]. Chen et al. assessed HIV content in cells with different levels of expression of FTO and ALKBH5 eraser proteins, finding that decreasing methylation levels enhanced binding to RIG-I [133].

### 5.2. Orthomyxoviruses

Influenza A virus (IAV) belongs to the *Orthomyxoviridae* family, characterized by negative-sense segmented RNA genomes that replicate in the host cell nucleus. Courtney et al. investigated how m^6^A affects the expression and replication of IAV genes. The authors demonstrated that METTL3 knockout and chemical inhibition of methylation using 3-deazaneplanocin A in A549 cells reduced IAV replication levels by suppressing viral mRNA and protein expression. The study also showed that increased regulation of the reader protein YTHDF2 significantly enhanced IAV replication and viral particle production. Additionally, the locations of m^6^A residues on viral mRNA were mapped using RNA–protein crosslinking and immunoprecipitation. These mapping data were confirmed by generating mutant forms of IAV, in which eight prominent m^6^A sites on the plus-strand mRNA/cRNA of hemagglutinin (HA) or nine m^6^A sites on the minus-strand vRNA of HA were mutated. In these IAV mutants, selectively lower levels of mRNA and HA protein were expressed, while the expression of other IAV mRNA and proteins remained unaffected [134]. Additionally, Zhu et al. identified the involvement of another reader protein, YTHDC1, which was found to be upregulated in IAV-infected cells. Using mass spectrometry and co-immunoprecipitation experiments, the authors showed that YTHDC1 interacts with the non-structural protein 1 (NS1) of IAV, thereby promoting viral replication. Mutations in NS1 (NS1 R38A and K41A) disrupted its ability to inhibit NS segment splicing and hindered its interaction with YTHDC1. Knockdown of YTHDC1 enhanced NS segment splicing, while restoring YTHDC1 reversed this effect. The study concluded that the interaction between NS1 and YTHDC1 plays a critical role in regulating NS mRNA splicing during IAV infection [11].

### 5.3. Flaviviruses

Flaviviruses are a group of viruses characterized by positive-sense ssRNA genomes that replicate in the cytoplasm. This viral family includes pathogens such as yellow fever virus, West Nile virus, dengue virus, tick-borne encephalitis virus, Zika virus (ZIKV), and hepatitis C virus (HCV). Lichinchi et al. observed that ZIKV can undergo m^6^A methylation mediated by host METTL3 and METTL14 as well as by ALKBH5 and FTO. m^6^A methylation was shown to negatively regulate ZIKV production. Knockdown of YTHDF1–3 readers increased viral RNA levels, while overexpressing YTHDF1-3 decreased levels of extracellular viral RNA [135]. Gokhale et al. conducted a comprehensive study mapping m^6^A modification sites in HCV, dengue virus, yellow fever virus, ZIKV, and West Nile virus. In addition to mapping, the authors investigated the role of m^6^A methylation in HCV and noted that depleting METTL3 and METTL14 increased the rate of HCV infection, promoting the formation of infectious viral particles without affecting viral RNA replication. Depleting the m^6^A demethylase FTO had the opposite effect. Additionally, YTHDF readers reduced the production of HCV particles and localized to viral assembly sites. All three YTHDF proteins bound to HCV RNA at the same sites, and their depletion increased the production of HCV particles [136]. The role of the YTHDC2 reader was established in the study by Kim and Siddiqui. Using cells with a knockout of the YTHDC2 helicase domain (YTHDC2-E332Q), they showed that the expression levels of HCV core protein were higher in cells transfected with wild-type YTHDC2 but significantly lower in cells transfected with YTHDC2-E332Q. Additionally, in the same study, the authors noted that suppressing METTL3 and METTL14 increased the stability of HCV RNA [137]. In experiments with HBV and HCV, Kim et al. showed that METTL3/METTL14 depletion led to increased viral RNA recognition by RIG-I with subsequent increases in IFN1 production. Furthermore, RNA binding by YTHDF2 and YTHDF3 was found to be the reason for decreased RIG-I sensing [138]. The mechanism behind the evasion of the RIG-I-mediated antiviral response through m^6^A methylation and mimicry of cellular RNAs has been identified in representatives of the *Pneumoviridae*, *Paramyxoviridae*, and *Rhabdoviridae* viral families [139,140]. 

Gokhale et al. found negative regulation of HCV viral particle production by METTL3/METTL14 methyltransferases and YTHDF1, YTHDF2, and YTHDF3 readers. Additionally, FTO demethylase positively regulated HCV particle production, while ALKBH5 levels did not affect viral particle production [136]. Further, m^6^A methylation of HCV RNA was shown to downregulate viral particle production and reduce the efficiency of RIG-I-mediated viral RNA sensing.

### 5.4. Coronaviruses

SARS-CoV-2, a β-coronavirus with positive-sense ssRNA, contains eight m^6^A modifications in its genome, as shown via m^6^A-seq and miCLIP analysis conducted by Liu et al. The authors also demonstrated that m^6^A RNA methylation negatively regulates the life cycle of SARS-CoV-2. In Huh7 cells infected with SARS-CoV-2, abundant METTL14 and ALKBH5 relocated to the cytoplasm, where coronavirus genomic RNA replication occurs. Virus replication and the percentage of SARS-CoV-2-positive cells significantly increased after METTL3 and METTL14 knockdown. Knocking down YTHDF2, but not YTHDF1 or YTHDF3, promoted viral infection and replication. SARS-CoV-2 can influence host cell m^6^A methylation, as the overall intensity of m^6^A significantly increased in Huh7 cells infected with SARS-CoV-2 compared to uninfected Huh7 cells [141]. Burgess et al. demonstrated that METTL3, YTHDF1, and YTHDF3 are necessary for the replication of human β-coronaviruses. The catalytic function of METTL3 plays a crucial role in the efficient synthesis of viral RNA within the first 24 h post infection, leading to subsequent accumulation of viral proteins [142]. Liu et al. conducted a systematic analysis of m^6^A in different strains of SARS-CoV-2 causing mild or severe forms of COVID-19, and concluded that the presence of more m^6^A modifications in the N region of SARS-CoV-2 correlated with weaker pathogenicity. The authors described several methylation sites that may be associated with viral pathogenicity, such as site 74 located in the transcription regulatory sequence of the 5′-UTR of SARS-CoV-2, and a similar site, 29 707, in the 3′-UTR [143].

### 5.5. Hepadnaviruses

HBV is a hepatotropic DNA virus from *Hepadnaviridae* family, characterized by a complex life cycle involving a stage of reverse transcription. Imam et al. used m^6^A-seq analysis to identify m^6^A methylation sites within a conserved motif located in the epsilon stem loop at the 3′ end of all HBV mRNA, as well as at the 5′ and 3′ ends of pregenomic RNA (pgRNA). Their results showed that m^6^A methylation in the 5′-epsilon stem loop of pgRNA was necessary for pgRNA reverse transcription, while m^6^A methylation in the 3′-epsilon stem loop of pgRNA destabilized all HBV transcripts [144]. Moreover, m^6^A RNA modification of HBV is necessary for efficient replication of the virus in hepatocytes. As shown by Murata et al., m^6^A modifications primarily occur in the coding region of HBx, and mutating m^6^A sites decreased HBV and HBs RNA levels. The authors also assessed the impact of the m^6^A methylation inhibitor cycloleucine on HBV. Cycloleucine reduced HBV RNA levels in cells and HBs levels in a dose-dependent manner, while 3-[4,5-dimethylthiazol-2-yl]-5-[3-carboxymethoxyphenyl]-2-[4-sulfophenyl]-2H-tetrazolium (MTS) analysis showed that cell viability was unaffected. METTL3 knockdown also reduced HBV and HBs RNA levels, indicating the crucial role of m^6^A in the HBV life cycle [145]. Kim et al. found that m^6^A methylation of HBV transcripts regulates their intracellular distribution. Transcripts of HBV in cells transfected with m^6^A mutant plasmids predominantly accumulated in the nucleus compared to wild-type cells. The authors also noted that knocking out YTHDC1 and fragile X mental retardation protein (FMRP) affected the nuclear export of m^6^A-modified HBV transcripts. MeRIP analysis showed that most m^6^A-modified viral RNAs located to the cytoplasm, but when cells are depleted of YTHDC1 or FMRP, m^6^A-methylated viral RNAs accumulated in the nucleus [146]. 

Studies by Imam et al. demonstrated that depleting METTL3 and METTL14 resulted in increased expression of HBV HBs and HBc proteins. Similarly, depleting FTO and ALKBH5 demethylases decreased expression of said proteins, suggesting that m^6^A methylation downregulates HBV protein expression. Notably, depleting YTHDF2 and YTHDF3 also resulted in increased expression of HBs and HBc proteins. Further investigations uncovered a more than two-fold increase in pgRNA stability in cells depleted of either YTHDF2 or METTL3/METTL14; this was ruled to be due to reduced stability of m^6^A-modified HBV RNA. This effect might be due to YTHDF2-mediated RNA degradation. The role of m^6^A methylation in the interaction between HBV and cells is further discussed by Kostyusheva et al. [147]. 

Zhang et al. conducted a bioinformatics analysis of m^6^A regulators associated with immune infiltration in HBV-related HCC. The authors demonstrated that m^6^A influences oncogenesis, tumor microenvironment, and patient prognosis in HBV-related HCC [148].

### 5.6. Adenoviruses

Adenoviruses (AdV), dsDNA viruses, undergo replication in the nucleus utilizing cellular RNA polymerase II. It was demonstrated that methyltransferases METTL3, METTL14, and WTAP, and the reader protein YTHDC1, translocate to sites of viral RNA biosynthesis in A549 cells infected with AdV5 within 18 h of infection, suggesting that m^6^A modification may play a role in regulating late viral RNA splicing efficiency. Knockout of METTL3 or WTAP did not affect the splicing efficiency of the E1A gene, expressed during early infection stages, but significantly reduced splicing efficiency of the fiber gene, expressed later. Similar changes were observed in cells lacking YTHDC1, albeit to a lesser extent [149]. Hajikhezri et al. noted that the absence of FXR1 reduced the accumulation of viral capsid protein during human AdV-5 infection in cell culture. The authors found that the FXR1 protein accumulates at a late stage of human AdV-5 infection and forms distinct subcellular condensates. CLIP-qPCR revealed that the endogenous FXR1 protein binds to m^6^A-modified viral capsid mRNA [150].

### 5.7. Herpesviruses

Transcripts of the HSV-1 genome can contain 12 sites of m^6^A methylation. Feng et al. noted that the expression of methyltransferases METTL3 and METTL14 and readers YTHDF1–3 increased in the early stages of HSV-1 infection and decreased in the later stages; the expression of demethylases FTO and ALKBH5 was reduced [151]. Hesser and Walsh also observed that the infection time of HSV1 suppresses YTHDF proteins [112]. Jansens et al. found that suppressing m^6^A-methylated HSV transcripts depended on the YTHDF reader protein family and correlated with the localization of these proteins to enlarged P-bodies [152]. Wang et al. identified that METTL3 is aberrantly expressed in mouse corneal endothelial cells and human umbilical vein endothelial cells infected with HSV1. HSV-1 infection may lead to increased levels of m^6^A in endothelial cells, but this does not always correlate with METTL3 levels [153]. Xu et al. demonstrated that the level of m^6^A RNA modification changed after infection of oral epithelial cells with HSV-1, likely regulated by changes in the expression of demethylases FTO and ALKBH5 [154].

## 6. Conclusions and Future Directions

Despite RNA methylation being described over forty years ago, the expanding significance of epitranscriptomic markers is still to be understood. m^6^A methylation is the most common and well-studied of these markers, but its role in virology was practically unstudied until recently. Now, viral epitranscriptomics has actively started to develop. Increasing attention is given to the role of m^6^A in regulating the reverse transcription and translation of viral genetic material, as well as the mechanisms of viral evasion of the host organism’s immune system. m^6^A modifications can determine the strength and duration of the activation of the innate immune response. Studying the impact of m^6^A methylation on innate antiviral immunity, viral life cycles, and the ability of viruses to mask and evade sensor cells holds numerous perspectives and practical applications.

Investigating how m^6^A methylation can influence the expression of genes involved in the innate immune response may provide a comprehensive answer to how m^6^A modifications regulate the transcription and translation of key genes associated with immunity.

Understanding the role of m^6^A modifications in the recognition of viral RNA by PRRs and the subsequent activation or inhibition of antiviral pathways warrants further investigation. Some factors recognizing viral DNA may be present not only in the cytosol but also in the nucleus [85]. This raises additional questions: how can proteins associated with innate immunity differentiate between host and viral DNA, and what role does m^6^A play in these processes?

Studying the functions of proteins linked to m^6^A methylation is also crucial. Since m^6^A is a dynamic regulation, understanding the role of the proteins regulating these modifications can provide an overall insight into the regulation of m^6^A modification during immune reactions and, consequently, the dynamism of immune responses.

Recently, the role of lncRNAs in the organization of intracellular immunity was actively investigated [155]. Exploring the influence of m^6^A methylation on lncRNAs associated with innate immunity will allow understanding of how m^6^A modifications in lncRNAs can affect their stability, localization, and interactions with other cellular components.

m^6^A is not the only modification to play a role in the organization of intercellular immunity [118]. Researching the interaction between m^6^A methylation and other epigenetic modifications in the context of innate immunity, such as how m^6^A interacts with DNA methylation, histone modifications, and other RNA modifications during the formation of immune responses, will establish additional mechanisms for regulating antiviral immune reactions.

An important direction is exploring the potential impact of m^6^A regulators as a therapeutic strategy for activating innate immune responses. Understanding regulatory mechanisms could open opportunities for developing interventions aimed at enhancing or suppressing immune reactions in various viral diseases. Studying how disrupting regulation of m^6^A modifications is linked to impairments in signaling pathways of antiviral immunity will help identify new diagnostic markers and therapeutic targets.

Investigating the involvement of m^6^A methylation in the innate immune response holds great prospects for further research into host–pathogen interactions and immune regulation. Studying m^6^A methylation’s role may have broad implications for the development of therapeutic agents related to modulating immune responses and treating viral diseases.

Recent studies have highlighted the potential of specific RNA modifications as valuable diagnostic and prognostic indicators in viral infections. For instance, Nagayoshi Y et al. reported an elevation in the levels of RNA modifications, including N^1^-methyladenosine (m^1^A), N^2^,N^2^-dimethylguanosine (m^22^G), N^6^-threonylcarbamoyladenosine (t^6^A), 2-methylthio-N^6^-threonylcarbamoyladenosine (ms^2^t^6^A), N^6^-methyl-N^6^-threonylcarbamoyladenosine (m^6^t^6^A), and N^6,2′^-O-dimethyladenosine (m^6^Am), following SARS-CoV-2 infection. 

The abundance of RNA modifications, specifically t^6^A and ms^2^t^6^A, exhibited a notable increase exceeding four-fold levels in response to SARS-CoV-2 infection. Through liquid chromatography–mass spectrometry (LC-MS) analysis of biological samples, the researchers proposed the potential utility of m^6^t^6^A and t^6^A as diagnostic biomarkers for COVID-19. Furthermore, the study demonstrated a positive correlation between the severity of COVID-19 and elevated levels of serum ms^2^t^6^A, suggesting the prognostic value of this RNA modification as a predictive marker for coronavirus infection [156]. 

Despite the abundance of data on the involvement of m^6^A in viral infections and other infectious diseases, there is a lack of comprehensive studies regarding their prognostic significance in terms of disease progression or therapeutic decision-making. In contrast, m^6^A has received considerable attention as a prognostic biomarker in oncology and other medical fields, facilitating the selection of optimal therapeutics, evaluation of disease progression risks, assessment of remission duration, and prediction of relapse risks.Additional investigation is warranted to advance the development of potential m^6^A-based prognostic biomarkers for infectious diseases. 

## Figures and Tables

**Figure 1 viruses-16-00601-f001:**
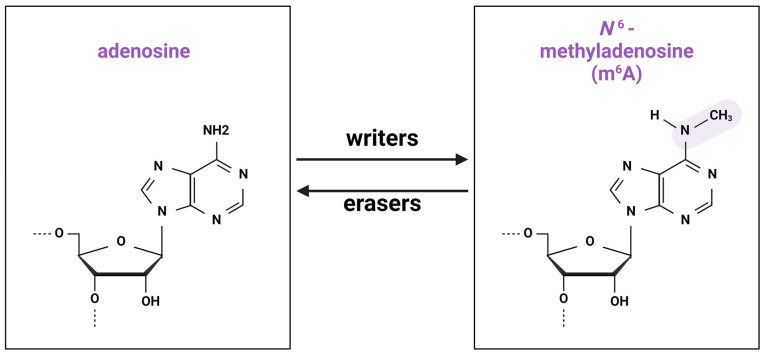
Chemical structure of m^6^A modification. Methylation and demethylation is implemented by groups of “writers” and “erasers”, correspondingly.

**Figure 2 viruses-16-00601-f002:**
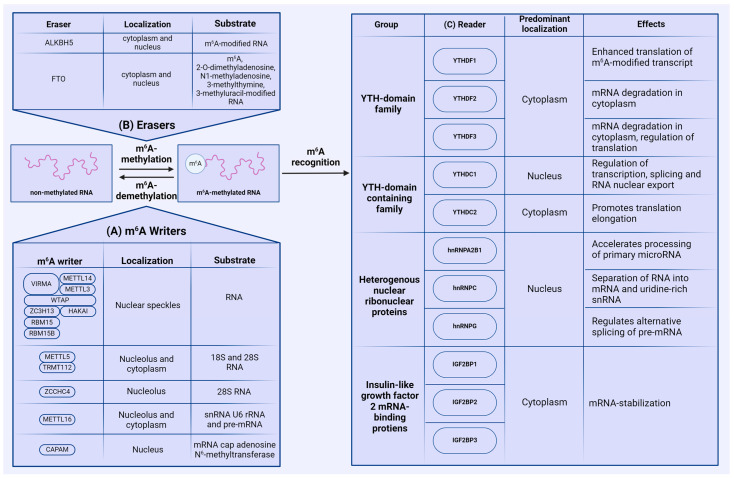
Mammalian m^6^A machinery. (**A**) RNA methylation is performed by groups of N^6^-methyltransferases (writers), with most of the m^6^A modification carried out by the methyltransferase complex, containing METTL3 and METTL14. (**B**) m^6^A demethylation is executed by demethylases (erasers) ALKBH5 and FTO. (**C**) m^6^A-methylated RNA is recognized by a plethora of reader proteins that play a crucial role in the fate of m^6^A-modified RNA.

**Figure 3 viruses-16-00601-f003:**
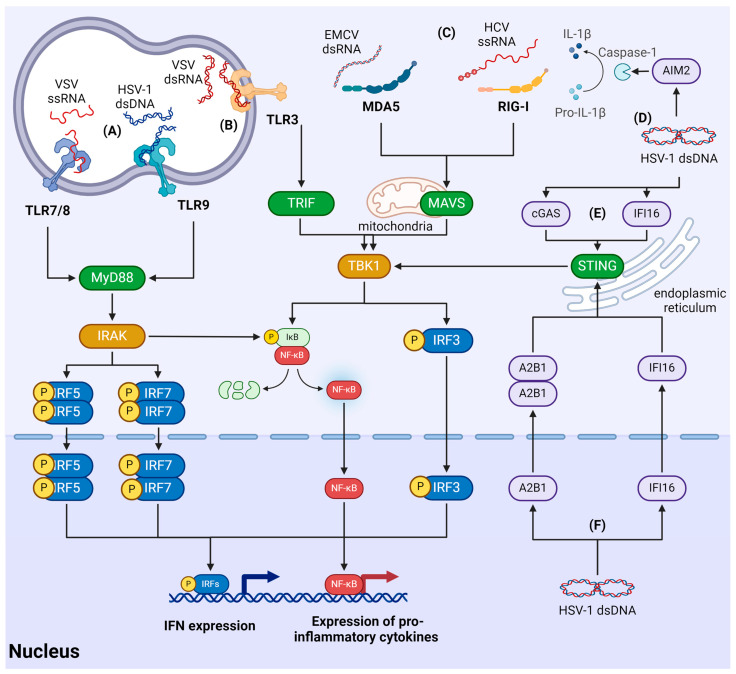
Antiviral immune response pathways. Endosomes: (**A**) ssRNA and CpG-unmethylated dsDNA are recognized by TLR 7/8 and TLR9, respectively, activating MyD88 that results in phosphorylation of IRF5, IRF7, and IκB by IRAK. Phosphorylated IRF5, IRF7, and NF-κB are then imported into the nucleus. IRF5 and IRF7 activate IFN expression, while NF-κB activates expression of pro-inflammatory cytokines. (**B**) dsRNA is sensed by TLR3, resulting in phosphorylation of IRF3 and IκB and their translocation into the nucleus. Ultimately, this pathway induces expression of IFN and pro-inflammatory cytokines. Cytoplasm: (**C**) viral dsRNA and ssRNA are sensed by MDA5 and RIG-I respectively. Both sensors activate MAVS and induce TBK1 leading to expression of IFN and pro-inflammatory cytokines via IRF3 and NF-κB. (**D**) Viral dsDNA is recognized by AIM2. AIM2 activates caspase-1, which mediates pro-IL-1β conversion into IL-1β. (**E**) dsDNA is also recognized by cGAS and IFI16, which activate STING, further inducing TBK1. In the nucleus. (**F**) dsDNA can be recognized by diverse DNA sensors, including hnRNPA2B1 (A2B1) and IFI16. Upon activation, these factors translocate to the cytoplasm and activate STING.

**Figure 4 viruses-16-00601-f004:**
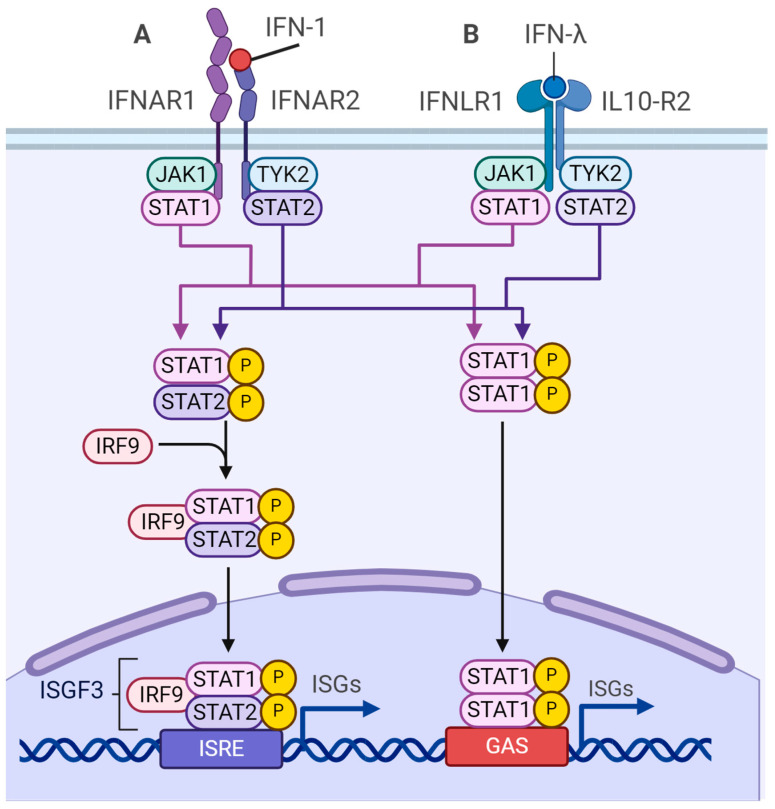
Cascades of type I and III IFNs signaling. Binding of IFN to its respective receptor triggers a signal transduction through activation of JAK1 and TYK2-mediated phosphorylation of STAT1 and STAT2, which translocate to the nucleus and stimulate ISGs expression. (**A**) IFN type I response. (**B**) IFN type III response.

**Figure 5 viruses-16-00601-f005:**
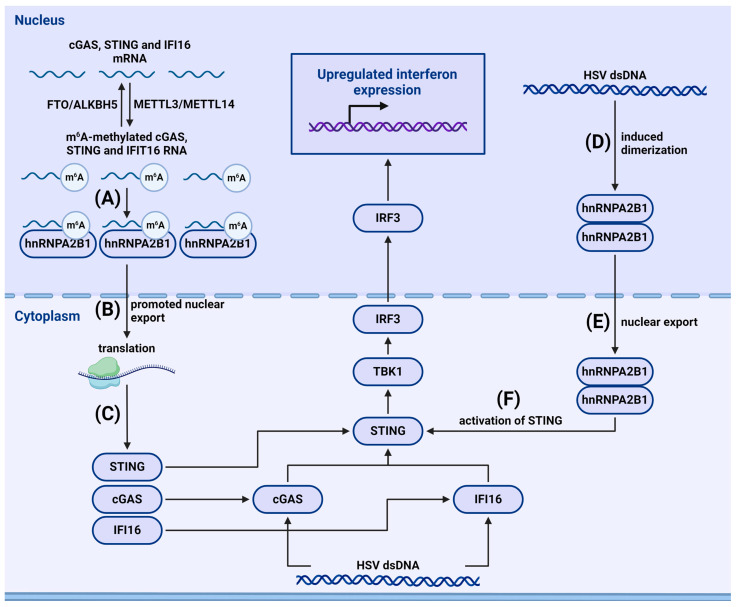
The role of hnRNPA2B1 in regulation of nuclear foreign DNA sensing. (**A**) Nuclear hnRNPA2B1 binds to m^6^A-methylated cGAS, STING, and IFI16 mRNA and (**B**) promotes their nuclear export. (**C**) Enhanced nuclear export of mRNA increases protein synthesis from mRNA of cGAS, STING, and IFI16, resulting in enhanced cytosolic foreign DNA sensing and antiviral signaling. (**D**) Homodimerization of hnRNPA2B1 in HSV infection. (**E**) Homodimerized hnRNPA2B1 exits the nucleus. (**F**) Activation of TBK1–IRF3 pathway via STING by hnRNPA2B1 in the cytoplasm.

**Figure 6 viruses-16-00601-f006:**
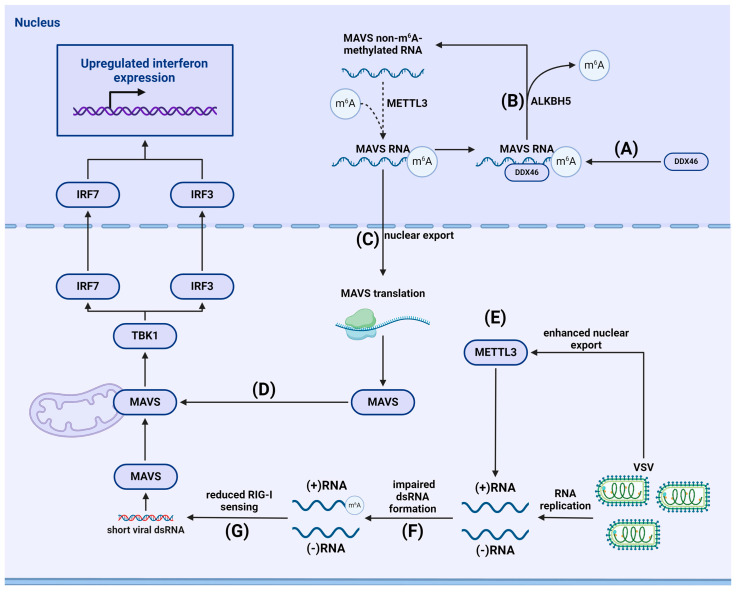
Interplay between VSV infection, m^6^A methylation, and RIG-I-dependent RNA sensing. (**A**) VSV causes DDX46 binding to m^6^A-methylated MAVS mRNA. (**B**) DDX46 recruits ALKBH5 and facilitates MAVS mRNA demethylation, (**C**) preventing its nuclear export. (**D**) Reduced nuclear export of MAVS mRNA leads to impaired RIG-I-mediated foreign RNA sensing. (**E**) VSV infection causes METTL3 translocation to the cytoplasm, where it methylates VSV (+) RNA. (**F**) m^6^A-methylation of VSV (+) RNA prevents dsRNA formation. (**G**) ssRNA is less prone to RIG-I sensing.

**Figure 7 viruses-16-00601-f007:**
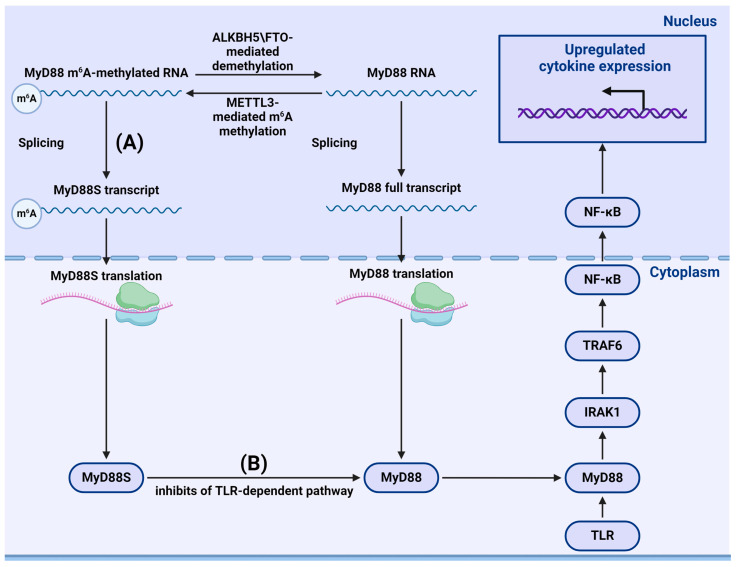
Impact of m6a methylation on TLR-dependent antigen sensing. (**A**) m^6^A methylation of MyD88 mRNA promotes alternative splicing and MyD88S transcript formation. (**B**) MyD88S protein prevents TLR-dependent signal transduction.

**Figure 8 viruses-16-00601-f008:**
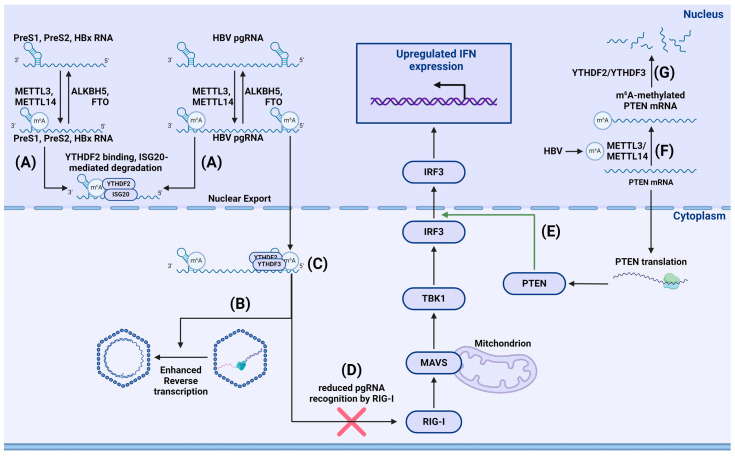
Impact of m^6^A methylation on RLR sensing of HBV. (**A**) m^6^a methylation of 3′ epsilon stem loop results in HBV RNA binding with YTHDF2 and subsequent ISG20-mediated degradation. (**B**) m^6^a methylation of 5′ epsilon stem loop enhances reverse transcription through currently unknown mechanism. (**C**) 5′ epsilon stem loop methylation causes binding of YTHDF2 and YTHDF3 to pgRNA. (**D**) YTHDF2/3 bound to pgRNA reduce pgRNA sensing by RIG-I. (**E**) PTEN enhances nuclear import of IRF3, leading to upregulated IFN production. (**F**) HBV infection increases m^6^A modification of PTEN mRNA. (**G**) Methylated PTEN mRNA is recognized by YTHDF2/YTHDF3 and subsequently degraded.

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
