# Peer review of "m6A Methylation in Regulation of Antiviral Innate Immunity"

_viruses, 2024, doi:10.3390/v16040601_

Round 1

Reviewer 1 Report

Comments and Suggestions for Authors

In this review, the authors stress the role of m6A in viral infections. The manuscript is clearly structured and well written. It is suggested that full name may be more appropriate instead of aberration, and some aberrations can be modified to become clear. The suggestions are as follows. Consequently, the reviewer recommends to publish the manuscript after the minor revision.

Suggestions:

  1. 114: “Cyclin A2 (CCNA2) and cyclin-dependent kinase 2 (CDK2)” would be better than “CCNA2 and CDK2” since those proper terms are used for the first time in this article.
  2. 128: “YTH521-B homology (YTH) domain” would be better than “YTH521-B domain (YTH)”
  3. 144: “Carbon catabolite repression 4-negative on TATA-less (CCR4-NOT)” would be better than “CCR4-NOT”
  4. 263: “DEAD-box RNA helicase 41 (DDX41)” would be better than “DDX41”
  5. 270: “Pyrin and hematopoietic interferon-inducible nuclear (HIN) domain (PYHIN)” would be better than “PYHIN”
  6. 278: “Apoptosis-associated speck-like protein containing a caspase recruitment domain (CARD) (ASC)” would be better than “ASC”
  7. 282: “DNA-dependent protein kinase (DNA-PK), oligoadenylate synthase (OAS)” would be better than “DNA-PK, OAS”
  8. 278: “Jumonji domain-containing 6 (JMJD6)” would be better than “JMJD6”
  9. 318: “C-X-C motif chemokine ligand 10 (CXCL10)” would be better than “CXCL10”
  10. 383: “Neuropilin-1 (NRP1), Tripartite motif containing 4 (TRIM4), Suppressor of Mothers against Decapentaplegic (SMAD) family member 4 (SMAD4)” would be better than “NRP1, TRIM4, SMAD4”
  11. 414: “Toll/Interleukin-1 receptor domain-containing adapter protein (TIRAP)” would be better than “TIRAP”
  12. 464: “phosphatase and tensin homolog (PTEN)” would be better than “PTEN”
  13. 487: “Forkhead box O3 (FOXO3)” would be better than “FOXO3”

    14. 540: “non-structural protein 1 (NS1) ” would be better than “non-structural    protein NS1”

Comments on the Quality of English Language

The manuscript is clearly structured and well written. 

Author Response

Please, see the responses to the Reivewers' critiques in the attached document. 

Reviewer 2 Report

Comments and Suggestions for Authors

The review from Karandashov et al. introduces the concept and recent advances in RNA m6A modification in regulation of antiviral innate immunity. m6A has been reported to play important roles in multiple physiological processes and disease states. And the current review focuses on how m6A posttranscriptional modification could interplay between viruses and host cells, what is immune responses and viral replication. This mechanism holds significant importance in the early drug discovery and development of drug candidates. Moreover, this mechanism paves the way for exploring the potential impact on m6A regulators as a therapeutic strategy for activating innate immune responses. Overall, this review provides valuable insights into the mechanism of m6A in the innate immune response. However, in order to enhance the manuscript, I believe there are some minor problems that the authors should address.

1.      In the first part, the author should introduce the dynamics of m6A in mammalian cells. It would be great if the authors could provide a figure to illustrate the processes.  

2.      At the end of the review, the authors briefly introduce the m6A modifications in viruses’ RNA. However, this part is crucial to induce the immunogenicity of host cells. The author should provide the mechanism of virus RNA modification. It should put at the beginning of the review, after the mechanism of m6A modification in mammalian cells. How viruses’ m6A modification is achieved and the dynamics is maintained.

3.      Even though, it’s a little bit off the focus of the topic, it would be great if the authors could mention the function of other RNA modifications in cellular immunogenicity.

4.      In the part three, the authors should also provide a figure to illustrate how the virus interact with five subfamilies of pathogen recognition receptors

5.      When they talked about the mechanisms of the antiviral innate immune responses, there are couple of key references are missing, for example, DOI: 10.1016/j.jmb.2013.11.024 and 10.3389/fimmu.2022.941931.

6.      There are multiple typos in the manuscript, for example, m6A in line 308, 422,473, 475.

Comments on the Quality of English Language

The english is fine, there is no misleading expression found in the current manuscript. 

Author Response

(The authors gave the same response as above.)

Reviewer 3 Report

Comments and Suggestions for Authors

The review paper by Karandashov et al. describes the current knowledges on how m6A is involved in viral replication and host immunity. 

Major points:

- The authors cover various preceding studies. However, the writing is a long list of current knowledges, and only the people who already know structured knowledge of the entire field can recognize the value of the manuscript. For newcomers to become able to understand the entire picture of the field, it is preferable for the authors to explicitly organize knowledges, describing the commonalities and differences of the various phenomena.

- The first figure does not appear until page 7. For a beginner of RNA modification to understand this review, it is preferable to add one figure in the section 2 “Regulation of m6A modification” on general things about m6A, e.g., chemical structure of m6A, the writers, readers, and erasers. For a beginner of immunology to understand this review, it is also preferable to add one figure in the section 3 “Mechanisms of the antiviral innate immune response” on general things on innate immune system, e.g., RLRs, TLRs, cGAS-STING, and their downstream pathways.

Minor points:

- Throughout the manuscript, the “6” within “m6A” describing the position of methylation should be written in superscript. 

-Page 1, line 27. Here in the Abstract, the authors write about “potential of m6A as a prognostic biomarker”, but there is almost no discussion on m6A as a biomarker. Of course, the studies on using m6A as a biomarker may be something for future studies, but there are already cases of other modified nucleosides (t6A and ms2t6A in blood or urine) that were recently found to be good biomarkers of coronavirus infection and severity (Nagayoshi et al. Biomolecules, 2022. PMID: 36139072), so maybe this preceding study can be used for discussion in the section 6 “Future directions”.

- Page 2, lines 67-69. The authors describe METTL3, METTL5, METTL16, and ZCCHC4 as the four known genuine m6A writers. But there are two more that add methyl group at the N6 position. One is CAPAM, a cap specific m6A writer for N6-methylation of the cap-specific m6Am modification (Akichika et al. Science. 2018. PMID: 30467178). Another is TRMO, which is a m6A writer for N6-methlation within m6t6A modification in the position 37 of specific tRNAs (Kimura et al. Nucleic Acids Research. 2014. PMID: 25063302). Although these two are not the major m6A writers, as the authors are focused on m6A, the authors should also lightly mention these two enzymes.

- Page 2, line 98. The “26S” should be written as “28S”?

Author Response

(The authors gave the same response as above.)

Round 2

Reviewer 3 Report

Comments and Suggestions for Authors

I think that this review paper is now a useful information resource for researchers and students who want to know about the role of m6A modifications in infection and immunity, with the following minor corrections.

- The '6' for m6A should be written in superscripts, 'm' in m6A in lower case and 'A' in m6A in capital letters in the following places: lines 29, 38, 44, 58, 63, 146, 405, 442, 451, 467, 468, 471, 484, 494, 521, 555, 572, 577, 599, 611, 634, 637, 675, 681, 687, 709, 716, 739, 746, 754, 776, 789, 790, 791, 807, and within Figure 5, Figure 6, Figure 7, and Figure 8.

- Similarly, the '5' for m5C (line 595), '4' for ac4C (line 595), '2' for ms2t6A (lines 818, 821, 826) and '6' for m6t6A and t6A (line 824) need to be written in superscripts.

- The '26S rRNA' in the new Figure 2 needs to be corrected to '28S rRNA'.

Author Response

We deeply appreciate the rigorous proofreading of the manuscript by the Reviewer. All required changes were introduced in the text and the Figures.